# Safe and Informed Use of Gadolinium-Based Contrast Agent in Body Magnetic Resonance Imaging: Where We Were and Where We Are

**DOI:** 10.3390/jcm13082193

**Published:** 2024-04-10

**Authors:** Francesca Iacobellis, Marco Di Serafino, Camilla Russo, Roberto Ronza, Martina Caruso, Giuseppina Dell’Aversano Orabona, Costanza Camillo, Vittorio Sabatino, Dario Grimaldi, Chiara Rinaldo, Luigi Barbuto, Francesco Verde, Giuliana Giacobbe, Maria Laura Schillirò, Enrico Scarano, Luigia Romano

**Affiliations:** 1Department of General and Emergency Radiology, “A. Cardarelli” Hospital, 80131 Naples, Italy; marco.diserafino@aocardarelli.it (M.D.S.); martina.caruso@aocardarelli.it (M.C.); giuseppina.dellaversanoorabona@aocardarelli.it (G.D.O.); costanza.camillo@aocardarelli.it (C.C.); vittorio.sabatino@aocardarelli.it (V.S.); dario.grimaldi@aocardarelli.it (D.G.); chiara.rinaldo@aocardarelli.it (C.R.); luigi.barbuto@aocardarelli.it (L.B.); francesco.verde@aocardarelli.it (F.V.); giuliana.giacobbe@aocardarelli.it (G.G.); marialaura.schilliro@aocardarelli.it (M.L.S.); luigia.romano@aocardarelli.it (L.R.); 2Neuroradiology Unit, Department of Neuroscience Santobono-Pausilipon Children’s Hospital, 80122 Naples, Italy; camilla_russo@hotmail.it; 3Department of Radiology, “San Carlo” Hospital, 85100 Potenza, Italy; enricoscarano@ospedalesancarlo.it

**Keywords:** magnetic resonance imaging, contrast medium, contrast media, gadolinium, adverse effect, gadolinium-based contrast agents (GBCAs), imaging sequences, MR cholangiography, hepatobiliary contrast agents, hepatobiliary imaging

## Abstract

Gadolinium-based contrast agents (GBCAs) have helped to improve the role of magnetic resonance imaging (MRI) for the diagnosis and treatment of diseases. There are currently nine different commercially available gadolinium-based contrast agents (GBCAs) that can be used for body MRI cases, and which are classifiable according to their structures (cyclic or linear) or biodistribution (extracellular-space agents, target/specific-agents, and blood-pool agents). The aim of this review is to illustrate the commercially available MRI contrast agents, their effect on imaging, and adverse reaction on the body, with the goal to lead to their proper selection in different clinical contexts. When we have to choose between the different GBCAs, we have to consider several factors: (1) safety and clinical impact; (2) biodistribution and diagnostic application; (3) higher relaxivity and better lesion detection; (4) higher stability and lower tissue deposit; (5) gadolinium dose/concentration and lower volume injection; (6) pulse sequences and protocol optimization; (7) higher contrast-to-noise ratio at 3.0 T than at 1.5 T. Knowing the patient’s clinical information, the relevant GBCAs properties and their effect on body MRI sequences are the key features to perform efficient and high-quality MRI examination.

## 1. Introduction

In magnetic resonance imaging (MRI), contrast agents are defined as drugs administered to temporarily change regional tissue properties in order to enhance the detection of possible abnormalities and improve anatomical depiction of organs and systems. The majority of MRI contrast agents is represented by chelates of the rare-earth Gadolinium. The free Gadolinium ion is highly toxic in vivo; therefore, a coordinating organic ligand is required to make it soluble, increase its stability and ensure safety and tolerance while it is administered [1,2]. These more complex compounds are also known as gadolinium-based contrast agents (GBCAs). Once administered (generally by intravenous injection) and distributed to target tissues/lesions, GBCAs are able to produce an increase in T1-weighted signals with an almost negligible effect on the T2-weighted signal; this property and the consequent ability to enhance tissue contrast are at the basis of their wide application in MRI examinations. Over time, several different types of GBCAs have been developed for clinical use, and some of them have progressively been withdrawn from the market due to safety issues. Conversely an ever-increasing number of new contrast media is under investigation, with particular reference to the ones intended to bind specific molecular targets and sense specific changes in pathological tissues; however, such new emerging agents, despite being promising, are still far from being approved for human use or implemented in daily clinical practice.

Here, we review some clue concepts concerning different types of approved GBCAs in order to guide the choice and optimize their use according to the specific diagnostic issue and to patients’ characteristics. In particular, in the first part of the paper we revise key core knowledge on gadolinium and GBCAs, required for an easier and more accessible reading of the subsequent sections. In the second part, we describe the most common strategies for GBCAs’ dose and contrast-enhanced imaging optimization. In the third part, we provide a comprehensive overview of the most common allergic and secondary reactions, as well as of the concerns related to GBCAs’ accumulation in human tissues. In the fourth part of the paper, we describe the indications and limitations in the use of GBCAs for specific at-risk categories, clarifying possible doubts and dispelling some myths.

## 2. Gadolinium and Gadolinium-Based Contrast Media: Key Concepts

The principal component of GBCAs’ structure is represented by gadolinium, a rare earth element, heavy metal, capable of attenuating X-rays (Figure 1), with the main property exploited for MRI being its high paramagnetic effect due to its seven unpaired electrons. Each electron creates a magnetic dipole that generates a local magnetic field, inducing increased relaxation in the adjacent water molecules.

Relaxivity is defined as the ability of a compound to increase the relaxation rates of the surrounding protons; gadolinium thus modifies longitudinal and transverse magnetic relaxation, shortening the T1 and T2 of the tissues where it concentrates. This T1 and T2 shortening results in an increase in the signal intensity on T1-weighted imaging and a reduction in T2-weighted sequences, with a prevailing T1-shortening effect at conventional doses; in case of higher gadolinium concentrations (for example, as it happens in the urinary bladder after contrast media kidney clearance), T2 shortening may result in a significant decrease in signal intensity noticeable on both T1- and T2-weighted images (with a possible “paradoxical” prevalence of T2 shortening on T1 shortening). Relaxivity and the subsequent effect on MRI contrast enhancement is primarily influenced by external factors such as temperature and field strength, and by molecular parameters including the hydration state of the molecule, substance in which the contrast agent is dissolved, or molecular size [3]. Concerning temperature, GBCAs are administered at room temperature and rapidly reach body temperature (37°) when injected; according to package inserts, no external heating is recommended to modify its viscosity and other molecular properties for routine clinical applications. Concerning the magnetic field, the T1-shortening effect due to contrast media is relatively poorly influenced by the magnetic field strength. Indeed, despite increasing the magnetic field, the relaxivity which is T1w GBCAs-related slightly decreases; the effect is globally compensated as the higher MRI field still offers an overall improved signal-to-noise ratio and contrast-to-noise ratio. As a result, the contrast-enhancing effect of GBCAs seems to be more pronounced even at 3T MRI [4,5] (Figure 2).

This effect, resulting in a higher contrast-to-noise and signal-to-noise ratio at 3.0 T compared to 1.5 T, find important applications in gadolinium-enhanced MR angiography, in which satisfying the image quality can also be achieved with lower GBCAs’ doses; however, such dose variation can be associated with a signal loss that is more pronounced in the venous system. Therefore, this off-label reduction must be carefully evaluated according to the specific diagnostic suspicion [6,7,8,9].

Concerning the molecular structure of gadolinium chelates, GBCAs can be classified according to the architectural framework of the organic chelating ligands into linear and macrocyclic; in the linear complexes, the gadolinium is attached either in the middle or at the end of the molecule, whereas in the macrocyclic complexes the gadolinium is in the center of a close ring structure (Figure 3).

The macroscopic structure influences GBCAs’ stability, with linear open-chain complexes more prone to Gadolinium ion dissociation and subsequent undesired toxic effects compared to cyclic close-chain molecules (see the paragraph “GBCA-related adverse reactions and secondary effects” for a more detailed discussion) [10]; macrocyclic derivatives are, therefore, well suited for MRI applications due to high stability in physiological media and a relatively fast water exchange rate. Another important property of GBCAs is ionicity, intended as the ability to dissolve into charged particles when entering a solution; this is strictly related to osmolality (number of dissolved particles/kg of water). However, the final osmolar effect is not determined by the contrast media concentration measured in the vial, but by the real concentration in the blood (which is related to osmolality and the volume administered). Indeed, in computed tomography (CT), at conventional doses, the clinical superiority of non-ionic iodinated contrast media compared to ionic ones in terms of renal safety and adverse reactions has been well established; conversely, in MRI, for GBCAs such a difference between ionic and non-ionic molecules is far less relevant or even negligible in terms of their systemic osmotic effects (a contingency mainly due to the small volumes usually administered for clinical purposes). The decision of which GBCA to use is mainly determined by clinical indication to MRI. Indeed, not all GBCAs have the same biodistribution and can be used for the same purpose. There are three main categories of GBCAs according to how contrast media travel in the human body and concentrates in specific tissues: extracellular space agent, hepatocyte-specific contrast agents, and blood-pool agents. 

*Extracellular space agents (ECSAs):* This category includes agents rapidly distributing within the extracellular space (vascular space plus interstitial space); these molecules are quickly eliminated by the kidneys (about 100% renal excretion). ECSAs are widely used for thoracic, abdominal, and/or pelvic MRI studies. This category encompasses molecules such as gadoterate meglumine, gadobutrol, gadopentetate dimeglumine, gadodiamide, gadoversetamide, gadoteridol, and gadopiclenol.*Hepatocyte-specific contrast agents (HSCAs):* Only two molecules are commercially available in this class (gadoxetate disodium and gadobenate dimeglumine); once intravenously injected, they undergo hepatocytes uptake. Their elimination is a combination of biliary and renal clearance (in particular, 50% of godoxetate disodium is excreted in the biliary system, thus with a shorter hepatocellular imaging window occurring approximately 20 min after injection and with a shorter total acquisition time compared to gadobenate dimeglumine of which just 5% is excreted in the biliary system). Due to their properties, HSCAs are mainly used for characterizing focal liver lesions, especially in chronic hepatopathies; off-label indications include bile duct imaging (both pre- or post-surgical or post-traumatic), gallbladder evaluation, and cystic duct obstructions [11,12,13].*Blood-pool agents (BPAs):* The only GBCA in this category is gadofosveset trisodium, which was discontinued after commercialization due to marketing reasons [14]; this contrast agent temporarily binds albumin, allowing the molecule to persist longer in the blood flow providing an almost selective vascular phase for up to 1 h from injection. This allows for a high-resolution three-dimensional MR angiography and MR venography; this GBCA has approximately five times the relaxivity of ECSA, which allows the first pass MR angiography to be performed with similar image quality as ECSAs but with a lower dose. Indications encompass aortoiliac occlusive disease, abdominal aortic aneurysm or dissection, pulmonary embolism, and vein thrombosis [15].

Hereafter, in Table 1, a brief classification of major GBCAs according to their structure and biodistribution is presented.

## 3. Gadolinium-Based Contrast Media Optimization

### 3.1. Dose and Concentration

As per any other drug, also for GBCAs, the basic rule is to use the minimum agent volume required to obtain a specific diagnostic goal. The standard volume calculation for GBCAs depends on the dose by weight (approved dose is reported in the package insert of each different GBCA), the patient’s weight and gadolinium molar concentration (variable and reported in the package insert of each different GBCA), according to the following simple formula: Volume (mL) = Dose (mmol/kg) × Weight (kg)/Concentration (mmol/mL).

It is generally allowed rounding of decimal milliliters up/down to the number closest to the unit or rounding down to the nearest vial size in many circumstances (i.e., to save opening a new vial); the only significant exception is represented by small GBCA volumes (i.e., in infants and children), in which the use of decimal doses according to the volume formula output is more strictly required to avoid over/underdose. Therefore, knowing the differences in concentrations and doses for each GBCA is essential to ensure the correct injection volume administration [16]; a summary of GBCAs’ concentrations and doses approved for clinical use is reported in Table 2.

For a critical analysis, among ECSAs it can be clearly inferred that higher concentration formulations (i.e., Gadobutrol 1 mmol/mL) result in a lower volume of GBCA administration, according to the above-mentioned formula [17]. The only significant exception to this axiom is represented by Gadopiclenol, a recently introduced macrocyclic non-ionic GBCA which is characterized by a much higher T1 relaxivity than other ECSAs; this allows for a lower dosage to be used, with a potential positive impact on the issue of Gadolinium deposition in human tissues [4]. Gadopiclenol, already approved by FDA for clinical use in adults and pediatric patients aged 2 years and older has also recently been approved by EMA for marketing and clinical use in Europe following the positive opinion of the Committee for Medicinal Products for Human Use [18,19,20]. 

A somehow similar concept concerning higher T1 relaxivity and subsequently recommended dose also applies to Gadofosveset trisodium and BPAs, whose sole formal indication was MR angiography for aortoiliac disease according to the FDA and whose production was recently discontinued because of poor sales [21].

Also, HSCAs have slightly higher T1 relaxivity compared to other GBCAs, hence a greater signal intensity enhancement on T1-weighted images with a relatively lower dose [11,22]. This observation underpins the change in dose recommendation for Gadobenate diemglumine provided by EMA (which approved Gadobenate diemglumine for hepatobiliary imaging only) and justified discrepancies between EMA and FDA guidelines (the latter do not envisage restriction in Gadobenate diemglumine clinical applications). Conversely, no dose discrepancy between EMA and FDA is related to Gadoxetate disodium, whose main indication is represented by hepatobiliary imaging (although with no formal restriction by FDA compared to EMA); in both cases, the lowest dose providing sufficient enhancement for diagnostic purposes (in the case in point, 0.025 mmol/kg) should be used.

### 3.2. Choice and Timing of Post-Contrast MRI Sequences

Postcontrast T1-weighted fat-suppressed 3D gradient-echo (3D GRE) is probably the most precious source of information among contrast-enhanced body MRI sequences. However, in order to optimize the contrast-to-noise signal and overall image quality for each GBCA, it is important to fully understand the proper timing for each GBCA category [12,13,23,24,25,26,27]:When using ECSAs, in the early arterial phase, arterial structures are enhanced, while in the late arterial phase, hypervascular tissues (including normal parenchymas such as the pancreas, spleen, or renal cortex) are visible. The venous phase allows for the best liver enhancement. In the delayed/equilibrium phase (occurring between 3 and 5 min from contrast injection), interstitial and extracellular spaces are finally enhanced (Figure 4).

When using HSCAs, arterial venous and equilibrium phases can be assimilated to the one observed in ECSAs, but due to the lower contrast dose and the early hepatocyte uptake in the venous phase, the result is worse and ultimately sub-optimal compared to ECSAs; conversely the most crucial information is obtained in the delayed hepatobiliary phase (about 20 min from injection with gadoxetate disodium and about 45 min with gadobenate dimeglumine for liver lesions depiction, or longer for bile ducts evaluation as an off-label application). Indeed, when MRI examination is performed to exclude a bile duct leak, further delayed images are required, and it is mandatory to wait until the contrast reaches the duodenum (up to 3 h for gadobenate dimeglumine) (Figure 5 and Figure 6).

*When using BPAs*, despite at present being withdrawn from the market, BPAs presented with a first-pass angiographic phase like ESCAs but offered a very long steady-state phase (up to 1 h from injection) to accurately depict blood vessels; timing should, however, be tuned according to the specific clinical indication for MRI examination (arteries vs. veins imaging).

As it is easy to imagine, contrast-enhanced body MRI scanning may require long time periods and multi-timing post-contrast acquisitions. To make the imaging process more efficient, it can be useful to put into practice all these strategies to allow earliest GBCA injection and to perform some additional pulse sequences in the time gap between arterial/venous and delayed post-contrast 3D GRE sequences. The type of pulse sequence to adopt relies on which GBCA has been used as well as on the effect of the specific contrast media on image degradation. Hereafter, we present some major considerations regarding the topic:Pulse sequences that may benefit from previous GBCA injection:−Two-dimensional radial slab MR cholangiopancreatography after ECSA, as gadolinium reduces the signal intensity of the kidneys and renal collecting systems, which may improve the visualization of the biliary tract and pancreatic ducts (Figure 7);−Short tau inversion recovery (STIR) after gadoxetate disodium, as contrast-induced T1 shortening of hepatic parenchyma causes suppression of background liver signal, accentuating contrast between normal liver and focal hepatic lesions;−Moderately T2-weighted fat-suppressed (reducing signal of kidneys and urinary system, and slightly reducing signal of other abdominal organs);−2D GRE series (increased signal intensity of blood vessels).Pulse sequences not significantly influenced by GBCA injection:−Balanced steady-state free-precession (Figure 8);−Diffusion-weighted images (paying attention to possible susceptibility artifacts from gadolinium in the urinary system).Pulse sequences negatively influenced by GBCA injection:−Dual GRE in-phase and out-of-phase (interferes with the evaluation of fatty liver or fat-containing lesions);−Two-dimensional radial slab MR cholangiopancreatography/high-resolution 3D MR cholangiopancreatography after HSCAs (the biliary excretion may darken the bile ducts and degrade biliary duct visualization, potentially rendering these images nondiagnostic);−STIR images after ECSA administration;−Single-shot fast-spin echo (SSFSE) heavily T2-weighted sequences (interference due to the T2 shortening effect of GBCA).

## 4. GBCA-Related Adverse Reactions and Secondary Effects

GBCAs are generally safe and well tolerated by most patients when injected. Acute adverse reactions due to MRI GBCAs have a lower incidence compared to the ones reported for CT iodinated contrast media, ranging from 0.01% to 2.4% in recent statistics [28,29]. Generally moderate and self-limiting, such adverse reactions are probably largely related to the high osmolality of these complex compounds and include coldness, pain at the injection site, itching, nausea, headache, dizziness, paresthesias; life-threatening events; and death from hypersensitivity reactions, although representing exceptional circumstances, are also possible [30,31,32]. Due to these concerns, researchers are systematically re-examining acute adverse reactions and medium/long-term side effects related to GBCAs administration. In this section, we provide a comprehensive overview of the most common allergic and secondary reactions to these drugs, and we examine the major concerns related to GBCAs accumulation in human tissues.

### 4.1. Local Undesired Events: Injection Site Reactions and GBCAs Extravasation

Some patients experiment with transient and harmless adverse reactions at the injection site, including pain, warmth, or coldness; generally, self-limiting and transitory, these sensations are mainly attributable to the injection technique as well as to uneasiness or anxiety felt by the patient (Lalli effect), and do not require further treatment [33]. Milder local adverse reactions such as a skin rash and hives are also occasionally reported, and only require single administration of an antihistamine drug [29,34]. GBCAs extravasation is another possible local undesired occurrence, which is usually self-limiting, and is also possible thanks to the sophisticated technologies of modern MRI-compatible contrast agent injectors; however, more serious complications such as compartment syndrome or tissue necrosis have also sporadically been reported [35]. Extravasation for GBCAs has a prevalence of about 0.045%, significantly lower compared to iodinated contrast media; this difference may be referred to a lower volume, lower injection rate, and more frequent resorting to hand injections, rather than by pharmacological or distribution differences [36]. For extravasation prevention, professionals must be up to date on extravasation management standard guidelines and be familiar with the most common procedures to apply in case of extravasation. Such procedures include the following (in this order): stopping the administration of intravenous drugs as first signs of extravasation occur; disconnecting the intravenous tube from the cannula; aspirating the remaining drug from the cannula; refrigerating the anatomical area involved by resorting to local thermal treatments, thus limiting drug dispersion in interstitial tissues; notifying the physician and reporting the undesired event in medical records; and resorting to surgical consultation in case of massive extravasation [37,38]

### 4.2. Systemic Adverse Events: Acute and Late Reactions

Systemic adverse events may be classified according to the timing of their occurrence in acute and late adverse reactions; acute adverse effects occur within 1 h from GBCA injection, while late adverse effects occur from 1 h to 1 week from administration. Acute adverse effects may be due to allergy-like reactions (not strictly IgE-mediated), hypersensitivity (IgE-mediated), or chemotoxicity. Acute adverse events are usually represented by mild phenomena such as skin rash, urticaria, itching, and erythema (allergy-like/hypersensitivity mechanisms, with an incidence ranging from 0.004–0.7%), or nausea, vomiting, paresthesias, headache, and dizziness (supposed direct chemotoxicity, self-limiting conditions with a described incidence of up to 2.4%). Acute severe and life-threating reactions are far less common, with an overall incidence ranging from 0.001% to 0.01%, whereas fatal reactions to GBCAs are only exceptionally reported; they include bronchospasm, laryngeal edema, hypotensive shock, respiratory arrest, and arrhythmia. Later reactions are generally milder than acute forms, and are mainly represented by a skin rash with erythema, swelling, and pruritus, as well as nausea, vomiting, headache, and fever (whose actual dependence on contrast media administration has to be elucidated) [39,40,41].

Adverse events incidence rises up to eight times in case of previous moderate-to-severe acute reactions to GBCAs; a mild increase in the overall risk has also been described for patients with personal history of asthma or atopy requiring medical treatment [41]. In these cases, it is required to determine whether the use of GBCAs during MRI examination is strictly necessary, or if an alternative diagnostic technique or a different GBCA can be envisaged. It must be remembered that, according to the most recent European Society of Urogenital Radiology (ESUR) guidelines [41], premedication is no longer recommended, as there is no evidence of its protective role and no scientific publication can confirm its efficacy in reducing the likelihood of such events in at-risk patients; however, according to the American College of Radiology (ACR) [42], prudential corticosteroid premedication prior to repeated contrast-enhanced studies with similar contrast media can still be suggested. In case of previous reactions to GBCAs, it is good practice to refer the patient to a specialist in drug allergy in order to check for evidence of a true allergy to GBCAs or cross-reactivity with other molecules, in order to minimize the risk of new reactions in case of re-administration [41]. Similarly to premedication, preventive fasting before contrast media administration is no longer recommended to prevent nausea, vomiting, or aspiration [43].

### 4.3. Nephrogenic Systemic Fibrosis

First described in the early 2000s, nephrogenic systemic fibrosis (NSF) is considered as a very late adverse reaction to GBCAs administration (sometimes classified as deposition-related phenomenon) which generally occurs in dialyzed patients or in patients with end-stage chronic kidney disease (CKD), and especially in cases of multiple GBCAs administrations over time. NSF symptoms’ onset is described as being within the days or months after injection with GBCAs and is characterized at first by systemic fibrosis in the skin and subcutaneous tissues with pruritus and thickening (described as indurate skin plaques and papules on extremities and trunk), followed by variable involvement of lungs, heart, esophagus, and skeletal muscles [44,45,46]. The exact etiological mechanism of NSF is still largely unknown: the most widely accepted hypothesis is that lower stability molecules such as linear GBCAs are more susceptible to a chemical phenomenon called dissociation-transmetallation (in which endogenous ions can replace gadolinium ions from the chelate), with the subsequent release of free toxic gadoliunium ions that may deposit in tissues and promote pathologic fibrosis; this mechanism only occurs if the elimination of GBCAs from the body through the kidneys is significantly delayed, thus in patients with impaired renal function or under dialytic treatment [1,2]. While linear GBCAs are more unstable and prone to transmetallation, macrocyclic chelates are far more stable and are, therefore, less susceptible to such dissociation and are safer in use. Patients with severe kidney failure (eGFR between 15 and 29 mL/min/1.73 m^2^; CDK stage 4) or end-stage CDK (CKD stage 5; eGFR < 15 mL/min/1.73 m^2^), as well as patients with acute kidney injury (AKI) superimposed or not to CDK represent at-risk outpatients. Because of these considerations, at present, laboratory testing of renal function including eGFR is not mandatory in low-risk patients and for cyclic chelates administration, while it is strictly recommended in high-risk patients such as those with a single kidney, who have had a kidney transplant and surgery, who have history of known renal cancer and multiple myeloma, and who have history of CDK or previous AKI [47]. For dialyzed patients, see the following section. No known prophylaxis is currently available to reduce the risk of NSF [48]; moreover, there is no scientific evidence demonstrating the preventive role of a low GBCA dose (lower than recommended for diagnostic purpose) in avoiding NSF [49].

Concerning the development of NSF after GBCAs administration, both ESUR (according to EMA guidelines) and ACR provided risk stratification of the most commonly available contrast media, but with the presence of some major differences between societies [41]; a panel to summarize these discrepancies is shown in Table 3.

In summary, cyclic GBCAs’ structure seems to prevent the breakdown between the ligand and the Gadolinium ion; therefore, the risk for NSF in patients undergoing standard doses’ administration can be considered negligible also in CKD patients. Indeed, prior risk stratification through clinical questionnaires and eGFR assessment is optional. High-risk linear molecules for NSF include gadodiamide, gadoversetamide, and gadopentetic acid, which have been suspended for intravenous use in Europe by EMA (while according to ACR guidelines they may still be used after a comprehensive patients’ stratification based on eGFR, remaining formerly contraindicated only if CKD is stage 4–5 and AKI). It must be noted that, according to EMA, gadopentetic acid (Magnevist^®^) can be used for arthrography MRI using intra-articular administration. Finally, intermediate-risk linear molecules for NSF include gadobenic acid and gadoxetic acid, which are approved in Europe by EMA for hepatobiliary imaging only due to their alternative hepatobiliary excretion pathway, but this is unlike in Europe, where according to ACR guidelines Gadobenic acid (MultiHance^®^) is considered safe for patients with chronic kidney disease and the preliminary assessment of renal function is considered optional prior to intravenous administration [50].

### 4.4. Accumulation in Human Tissues

In recent years, an ever-increasing number of in vivo and ex vivo studies provided evidence of gadolinium retention in normal tissues after repeated GBCAs administrations, occurring with both linear and cyclic contrast agents despite having a different magnitude (greater with linear GBCAs than with macrocyclic GBCAs, probably due to the less labile structure of the latter). This aberrant and unexpected deposition occurs in patients regardless of preserved renal and hepatic function. Because of this evidence of dose-dependent accumulation after repetitive GBCAs administrations, caution is still warranted especially for linear GBCAs [51,52]; however, gadolinium deposition still is a relatively undefined phenomenon from a clinical perspective. Due to this evidence and to their at least in part undetermined clinical meaning, the EMA suspended from the European market some linear GBCAs due to concerns regarding gadolinium retention (Optimark^®^, Omniscan^®^) or severely restricted their diagnostic applications (Magnevist^®^, Eovist^®^, Primovist^®^ and MultiHance^®^) [50]. In the wake of this, the FDA published a safety alert stating the need to continue analyzing the risk and clinical significance connected to gadolinium deposits, but have not yet foreseen any suspension in the GBCAs market in the United States due to a lack of association between adverse events and gadolinium retention [53].

Gadolinium retention in bones, liver, and skin has been assessed with histology, as well as the one observed in the kidney (specifically in patients with NSF), but it cannot be detected using MRI. Gadolinium retention in brain tissue, although confirmed in several post-mortem studies, can also be observed at MRI examinations as a focal T1-weighted hyperintensity in specific central nervous system (CNS) regions [51,54,55]. Among the above-mentioned deposition sites, the most striking and groundbreaking reports first concerned gadolinium retention in CNS [56,57], the only one that can be observed in vivo by means of MRI examination as spontaneous unenhanced T1-weighted hyperintensity in the dentate nuclei and globus pallidus (Figure 9). The phenomenon is not limited to patients with blood–brain–barrier disruption; it is apparently dose-dependent and is more likely to occur with linear rather than with cyclic GBCAs [51]. 

The exact mechanisms by which gadolinium administration causes high signal intensity in these CNS regions remain unclear; however, at present there is no evidence supporting neurotoxic effects of such gadolinium depositions in the short and medium term, neither in animal models nor in humans [58,59,60,61]. However, the radiology community is still making a systematic effort with GBCAs exposure analysis to definitively assess safety, define potential undiscovered neurotoxicity, and modify guidelines accordingly as new evidence is collected.

## 5. Gadolinium-Based Contrast Agents and at-Risk Categories

Some clinical situations bring into question the safety profile of GBCAs and represent a potential source of application error for radiologists. Here, we revise some cardinal concepts for safe GBCAs’ use in specific at-risk outpatients, clarifying possible doubts and dispelling myths.

### 5.1. GBCAs and Dialyzed Patients

As per other patients with known or highly suspected kidney function impairment, in dialyzed patients, the use of low-risk NSF GBCAs is recommended. Radiologists are requested to always consider alternative equivalent examinations, to use the smallest contrast dose necessary for diagnostic purposes, and to avoid close re-administrations (at least 7 days between two injections) [41]. Indeed, dialysis effectively removes circulating GBCAs (up to 95% in three dialytic sessions); therefore, it is important to synchronize the timing of contrast agent administration with the timing of the scheduled dialysis session in patients with CKD who are already undergoing dialysis. At present, there is no evidence of the protective impact of prior prophylactic dialysis on reducing the risk of NSF or AKI [62,63], and contrast-enhanced MRI should be scheduled before a regularly programmed dialysis session. In case of urgent non-scheduled examinations, an extra dialysis session after contrast media injection can also be recommended; however, no consensus on this point has already been reached between the major scientific societies, and ACR committee still recommend not to alter dialysis timing in patients receiving low-risk NSF GBCAs [42].

### 5.2. GBCAs and Patients with Sickle Cell Disease

Despite some historical alleged facts and fallacies related to the risk increase in acute crisis after GBCAs administration in patients with sickle cell disease (SCD), there is no evidence that intra-venous GBCAs induce vaso-occlusive or hemolytic events when administered to SCD patients [64]. As if proving this point, in several studies on SCD patients GBCAs were administered for contrast-enhanced MRI examinations without reported adverse effects [65,66]. Therefore, no restriction in GBCAs administration must currently be envisaged solely on the basis of sickle cell trait or SCD, and no specific premedication is required.

### 5.3. GBCAs and Interaction with Other Drugs

Among the molecules requiring additional consideration, particular mention should be made of drugs that could interact with GBCAs or potentially enhance contrast-induced nephrotoxicity and renal adverse reactions; the list should specifically include metformin treatment for diabetes, non-steroidal anti-inflammatories, and chemotherapies. Concerning antidiabetic medications, it has been clarified that no special precaution is necessary for diabetic patients on metformin treatment, for patients under non-steroidal anti-inflammatory drugs, and for patients on Cyclosporine/Cisplatin treatment, given the low risk of AKI with GBCAs (provided that the renal function is preserved); therefore, stopping nephrotoxic drugs before GBCAs is not generally required. Concerning interleukin-2, patients with known previous GBCAs-related reactions or on interleukin-2 treatment should be informed of the remote chance of late skin reactions; radiologist should recommend contacting a specialist in case of cutaneous symptoms onset [41,67].

Another important point is the one concerning interactions with other contrast media, especially CT iodinated contrast media. Comprising 75% of both gadolinium- and iodine-based contrast agents excreted within the first 4 h from intravenous administration, the second injection can be performed from 4 h from the first diagnostic procedure; conversely and as previously stated, in case of subjects with renal function impairment, there should be an interval of 7 days between the two administrations. As a further annotation, when choosing in which order to perform contrast-enhanced CT and MRI, it must be considered that GBCAs significantly attenuate X-rays when excreted in the urinary and in the biliary tract (Figure 1) (therefore, abdominal CT should be performed before MRI); no order concerns are described in cases of chest or head-and-neck examinations [41].

Finally, GBCAs should not be mixed with other drugs before intravascular injection, as it may interfere with their molecular stability; moreover, it is good practice not to use for GBCAs injection the same venous access used for other drugs administration or (when this is the case) flushing the catheter with normal saline before and after contrast-media administration [67].

### 5.4. GBCAs during Pregnancy

Only a few studies evaluated the correlation between GBCAs administration during pregnancy and the harmful effects to the fetus or newborn, but at present no mutagenic or teratogenic effect was described. In animal models, accumulation of GBCAs in amniotic fluid is demonstrated until 2 h, despite having no clear clinical repercussions [68]. In humans, the most important evidence on this concern is represented by a retrospective review of a large database of newborns [69], in which no significant difference in the risk of congenital abnormalities was found between women having undergone contrast-enhanced MRI examination with GBCAs and those who have not; conversely, in newborns exposed to GBCAs in utero (independent from gestational age at the time of the MRI) an increased risk of a wide spectrum of rheumatological, inflammatory, or infiltrative skin conditions was observed, coupled with an increased risk of stillbirth or neonatal death. These conclusions determined the indication of ACR that GBCAs should be administrated with caution during pregnancy or in possibly pregnant women, and only if a potential critical benefit to the patient or fetus can justify the currently still-unknown risk of fetal exposure [42]. Similarly, ESUR allows the use of the smallest quantity of GBCA during pregnancy only in cases of very strong clinical indications of the contrast-enhanced MRI examination [41]. Both guidelines formerly recommend the use of one of the macrocyclic GBCAs at low-risk for the NSF. However, it must be noticed that the use of GBCAs during pregnancy, despite being rare, is far more common during the first weeks of gestation before pregnancy is known and sometimes even when pregnancy tests are still negative before embryo implantation in the second week, but, as the embryo is not yet directly exposed to circulating GBCAs, this inadvertent administration is not likely to result in gadolinium retention in the embryo, with possible mitigation of the potential harmful effects [70]. Finally, in pregnant women with known renal impairment, the use of GBCAs is formally contraindicated in Europe [41], while according to ACR the same safety measures as in non-pregnant women should be adopted.

A summary of the different guidelines between ESUR and ACR concerning the use of GBCAs in pregnant women is shown in Table 4.

### 5.5. GBCAs in Women Who Are Breast-Feeding

Women with preserved renal function excrete less than a 0.04% GBCA dose into breast milk within the first 24 h after intravenous injection, and infants absorb less than 1% of the swallowed drug through the gastrointestinal tract; therefore, the global dose absorbed by the infant is lower than 0.0004% of the intravascular dose administered to the mother and the connected risks for the infant can be considered negligible. This suggests that no breast-feeding interruption is formally required after GBCAs administration [71]. However, if the woman is particularly concerned about the potential undesired events for the infant, breast-feeding abstention can be proposed within the first 24 h from GBCAs administration to the mother and breast milk elimination from both breasts should be suggested during the same period. For scheduled MRI examinations, a breast pump to obtain milk before GBCA injection can be used to feed the infant during the abstention period. Abstention from breast-feeding does not need to take place beyond 24 h from GBCAs administration [42]. As a final remark, one must always bear in mind that in lactating women with known renal impairment the use of GBCAs is formally contraindicated [41].

A summary of the different guidelines between ESUR and ACR concerning the use of GBCAs in lactating women is shown in Table 4.

## 6. GBCAs in Children

At present, scientific evidence suggests that GBCAs are usually well tolerated by children and that the risk of adverse effects is apparently comparable to that observed in adults. Many adverse reactions are mild or moderate, and generally they are self-limiting with no need for therapy or hospitalization [72]. Considering pediatric life expectancy and the lack of studies on long-term GBCAs effects in children, it is of the utmost importance to weigh the risks and benefits of single or repeated contrast media administrations, so, when the diagnostic advantage deriving from contrast-enhanced MRI in children is significant, GBCAs can be used if the dose is adjusted for the patient’s age and weight [73]. However, GBCAs are mostly used off-label in children, and intermediate/high-risk agents should be avoided. Moreover, as several of these agents are still not approved for pediatric use, GBCA’s leaflet should be preliminarily consulted, and (when not formally approved GBCA is available) prior informed consent for off-label use must be obtained from parents. When absolute contraindication to the use of a specific GBCA in the pediatric population is reported in the leaflet, its use in children is formally proscribed regardless from parents’ consent [41,42]. No clear evidence for nephrotoxicity in children after GBCAs administration at approved doses has been described, and similarly there are only a few isolated reports of NSF in children [74]. As per adults, pediatric patients at risk for AKI/CKD should be identified; in these cases, age-specific normal parameters for assessing renal function must be measured, remembering to use a revised Schwartz equation to determine eGFR [32,33]: eGFR (mL/min/1.73 m^2^) = 36.5 × length (in cm)/serum Cr (in μmol/L).

## 7. Conclusions

Despite the efforts invested in the effective development of GBCAs for in vivo human imaging, only a handful of compounds gained the approval for current applications in clinical practice. To date, approved GBCAs are widely used in MRI examinations as positive contrast agents, in order to enhance detection of possible abnormalities and improve anatomical depiction of organs and systems. However, as for any other drug, GBCAs use is not without risk, and the long-term effects have long been under the magnifying glass for their potential clinical implications that have yet to be fully explored and interpreted. Therefore, a thorough knowledge of GBCAs’ properties as well as a deep understanding of their indications and limitations are strongly desirable to optimize their use, improve tolerance, avoid possible pitfalls, and minimize the risk of adverse effects.

## Figures and Tables

**Figure 1 jcm-13-02193-f001:**
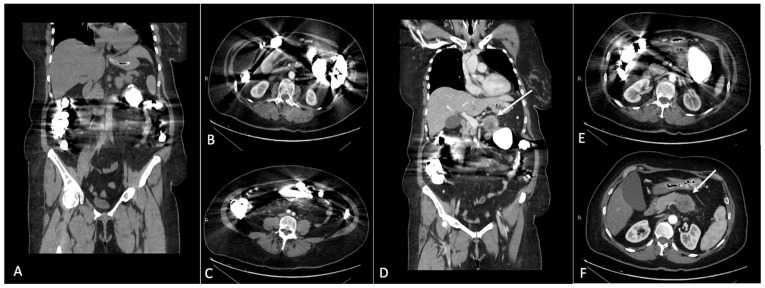
(**A**–**C**) Coronal and axial CT images after iodinated contrast medium intravenous injection in a patient with gadolinium accumulation in the bowel loops. Images show the streak artifacts generated by the presence of gadolinium contrast medium in the lumen of intestinal loops. (**D**–**F**) Please note the decreased imaging resolution of the pancreatic region. In (**D**,**F**), it is possible to identify a pancreatic solid mass in the body of the pancreas (arrows); the presence of the artifacts related to gadolinium in the bowel loops makes the evaluation of the pancreatic mass suboptimal.

**Figure 2 jcm-13-02193-f002:**
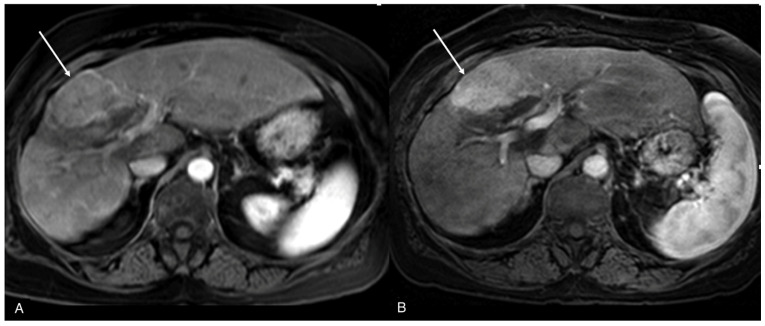
MRI of the same focal liver lesion ((**A**,**B**), arrows) on post-contrast T1w image at 1.5 T (**A**) and 3 T (**B**).

**Figure 3 jcm-13-02193-f003:**
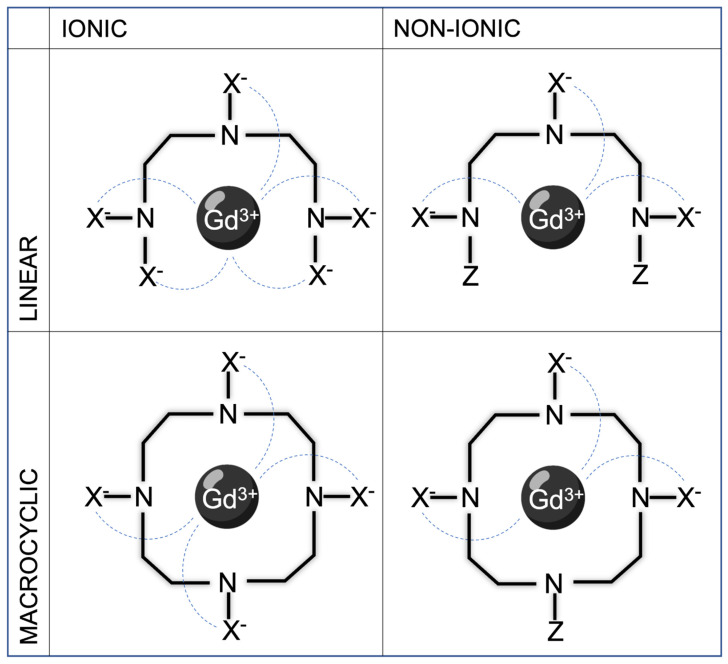
Examples of linear ionic, linear non-ionic, macrocyclic ionic, and macrocyclic non-ionic structure of GBCAs. Continuous lines represent single covalent bonds between atoms, while dashed lines represent ionic bonds due to weaker electrical attraction.

**Figure 4 jcm-13-02193-f004:**
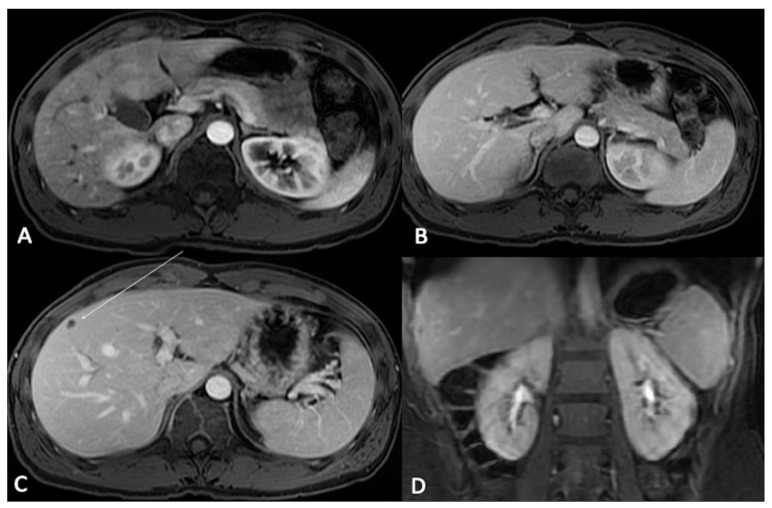
T1-weighted Liver Acquisition Volume Acceleration (LAVA) axial (**A**–**C**) and coronal (**D**) images after ECSA intravenous administration in the different phases of a complete dynamic post-contrast RM study. (**A**) The early arterial phase shows the arterial structures, generally represented by arterial vascular structures; (**B**) the late arterial phase shows the enhancement of hypervascular tissues such as abdominal parenchymas: please note that it is possible to evaluate pancreas, spleen, or renal cortex. (**C**) In the venous phase, the liver enhancement is optimally visualized: this phase represent the best moment to evaluate the liver parenchyma and focal lesions, as demonstrated by the clear visualization of a small hypointense focality in the subcapsular plane (arrow). (**D**) In the delayed/equilibrium phase, interstitial and extracellular spaces are finally enhanced, and it is possible to evaluate urinary excretion of contrast and opacification of renal collecting system.

**Figure 5 jcm-13-02193-f005:**
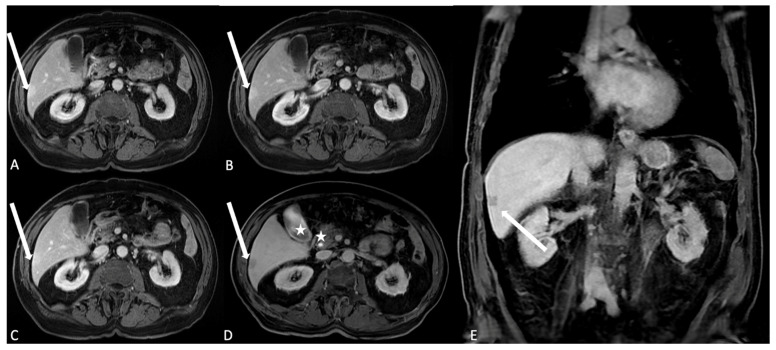
T1 LAVA axial images after HSCA (gadobenate dimeglumine) intravenous administration. (**A**) The arterial phase. Arterial structures and hypervascular lesions are evidenced: in the liver segment 6, it is possible to identify a subcapsular hyperintense area as indication of a hypervascular behavior (arrow); (**B**) in the venous phase, the liver parenchyma reach the best enhancement and the hypervascular area in segment 6 shows persistent enhancement (arrow); (**C**) the delayed/equilibrium phase allows representation of interstitial and extracellular spaces enhancement. The subcapsular lesion is quite completely homogeneous to the liver parenchyma arrow, suggesting the angiomatous nature of the lesion. (**D**,**E**) Axial and coronal images after HSCA (Multihance) intravenous administration of the same patient acquired in the delayed hepatobiliary phase show the opacification of the gallbladder lumen and the choledocic duct (stars); the subcapsular lesion is also identified as the hypointense area (arrows), confirming the vascular nature of the angioma and the lack of hepatocyte uptake and biliary excretion.

**Figure 6 jcm-13-02193-f006:**
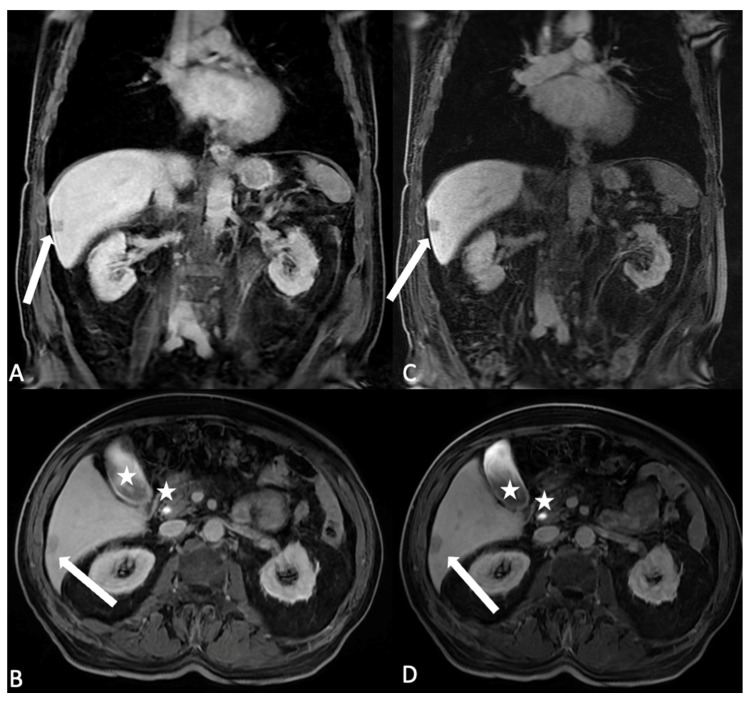
Axial and coronal T1 LAVA images after HSCA intravenous administration (gadobenate dimeglumine) of the same patient in hepatobiliary phase with different flip angle (FA) settings. (**A**,**B**) The FA is settled at 10 degree: the liver parenchyma results in the hepatobiliary excretion phase, and it is possible to identify the lesion with no hepatocyte uptake as hypointense (arrow), (**C**,**D**) changing the FA to 30 degree the hypointensity of the non hepatocitary lesion is better visible and identification is easier (arrow); moreover, the biliary signal results in better appreciated hyperintense signal (stars). Please note in the image, with the FA settled to 30 degree, the noise of the images increases; the modified flip angle is planned just in the last phases of acquisition.

**Figure 7 jcm-13-02193-f007:**
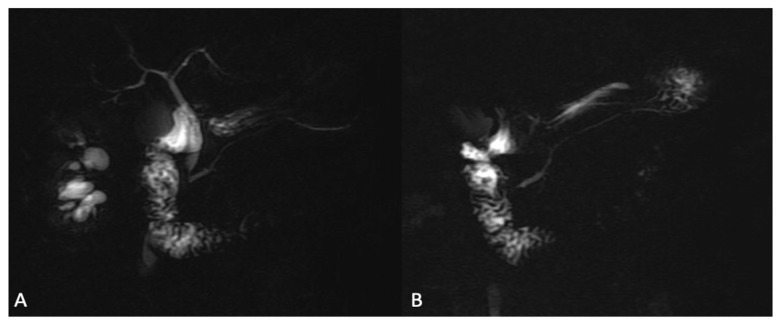
Images of 2D radial slab MR cholangiopancreatography. (**A**) The 2D cholangiopancreatographic image shows the anatomy of biliary tree and of the gallbladder with a hyperintense signal derived by biliary fluid. (**B**) The 2D cholangiopancreatographic image is acquired after ESCA injection, showing a better imaging representation with reduced intensity signal of the kidneys and renal collecting systems, which may improve the visualization of the biliary tract and pancreatic ducts.

**Figure 8 jcm-13-02193-f008:**
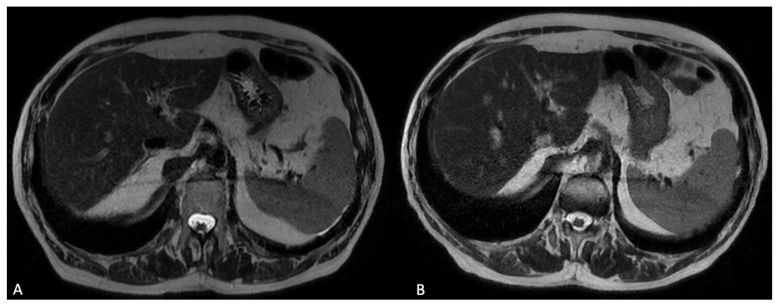
Axial T2 weighted images before (**A**) and after (**B**) HCSA (gadobenate dimeglumine) intravenous administration; no significative differences in signal intensity are visible; and T2 weighted images may be acquired after HSCA administration in order to reduce the timing of the acquisition protocol.

**Figure 9 jcm-13-02193-f009:**
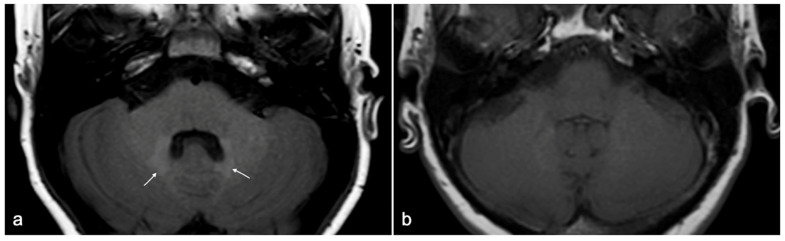
(**a**) Example of spontaneous T1w hyperintensity due to gadolinium deposition within dentate nuclei (white arrows) at 1.5 T MRI in a 25-year-old patient diagnosed with tuberous sclerosis who has undergone multiple Gadobutrol administrations over years. (**b**) Age- and sex-matched normal control for comparison.

**Table 1 jcm-13-02193-t001:** Schematic representation of molecular structure, ionicity, and biodistribution of GBCAs, as well as approval according to European Medicines Agency (EMA) and Food and Drug Administration (FDA).

Active Principle	Commercial	Manufacturer	Structure	Charge	Distribution	EMA	FDA
Gadoterate meglumine	Dotarem^®^ Clariscan^®^	Guerbet, Aulnay-sous-Bois, France	Cyclic	Ionic	ECSA	Approved	Approved
Gadobutrol	Gadavist^®^ Gadovist^®^	Bayer, Milano, Italy	Cyclic	Non-Ionic	ECSA	Approved	Approved
Gadopentetate dimeglumine	Magnevist^®^	Bayer Pharma, Leverkusen, Germany	Linear	Ionic	ECSA	Suspended	Approved
Gadodiamide	Omniscan^®^	GE Healthcare, Milano, Italy	Linear	Non-Ionic	ECSA	Suspended	Approved
Gadoversetamide	Optimark^®^	Mallinckrodt Deutschland GmbH, Hennef, Germany	Linear	Non-Ionic	ECSA	Suspended	Approved
Gadoteridol	ProHance^®^	Bracco, Hennef, Germany	Cyclic	Non-Ionic	ECSA	Approved	Approved
Gadopiclenol	Vueway^®^ Elucirem^®^	Bracco, Raleigh, NC, USA	Cyclic	Non-Ionic	ECSA	Approved	Approved
Gadoxetate disodium	Eovist^®^ Primovist^®^	Bayer, Milano, Italy	Linear	Ionic	HSCA	Restricted	Approved
Gadobenate diemglumine	MultiHance^®^	Bracco, Colleretto, Italy	Linear	Ionic	HSCA	Restricted	Approved
Gadofosveset trisodium	Ablavar^®^ Vasovist^®^	Bayer Pharma, Berlin, Germany	Linear	Ionic	BPA	Withdrawn	Withdrawn

**Table 2 jcm-13-02193-t002:** Summary of concentration (mmol/mL), dose (mmol/kg), and T1 relaxivity at 1.5 T (L/mmol-s) of GBCAs according to their package inserts and to EMA/FDA guidelines (see notes).

Type	Active Principle	Commercial	Concentration	Dose mmol/kg	Dose mL/kg	T1 rel	Notes
ECSA	Gadoterate meglumine	Dotarem^®^ Clariscan^®^	0.5 M	0.1	0.2	3.6	Adults and pediatric patients (including term neonates).
	Gadobutrol	Gadavist^®^ Gadovist^®^	1 M	0.1	0.1	5.2	Adults and pediatric patients (including term neonates).
	Gadopentetate dimeglumine	Magnevist^®^	0.5 M	0.1	0.2	4.1	Suspended by EMA for intravenous use. According to FDA: adults and pediatric patients (including term neonates).
	Gadodiamide	Omniscan^®^	0.5 M	0.1	0.2	4.3	Suspended by EMA. According to FDA: adults and pediatric patients aged 2 years and older; for imaging the kidney, halving the dose (0.05 mmol/kg) is recommended.
	Gadoversetamide	Optimark^®^	0.5 M	0.1	0.2	4.7	Suspended by EMA. According to FDA: contraindicated up to 4w, not recommended up to 2 y of age.
	Gadoteridol	ProHance^®^	0.5 M	0.1	0.2	4.1	Adults and pediatric patients (including term neonates). According to FDA: supplementary dose (0.2 mmol/kg) may be given up to 30 min after the first dose in adults without renal impairment if poorly visualized CNS lesions or equivocal MR scan.
	Gadopiclenol	Vueway^®^ Elucirem^®^	0.5 M	0.05	0.1	12.8	Approved by EMA and FDA: adults and pediatric patients aged 2 y and older.
HSCA	Gadoxetate disodium	Eovist^®^ Primovist^®^	0.25 M	0.025	0.1	6.9	Not recommended for use in children below 18 y. According to EMA: approved for hepatobiliary imaging only; organ-specific imaging of liver at 0.025 mmol/kg. According to FDA: allowed up to 0.05 mmol/kg, but at present recommended at 0.025 mmol/kg for hepatobiliary imaging only.
	Gadobenate diemglumine	MultiHance^®^	0.5 M	0.05–0.1	0.1	6.3	According to EMA: approved for hepatobiliary imaging only; organ-specific imaging of liver at 0.05 mmol/kg. According to FDA: no restriction (i.e., also indicated for CNS imaging and MR angiography); recommended dose 0.1 mmol/kg in adults and pediatric patients aged 2 y and older; halving the dose in pediatric patients aged less than 2 y.
BPA	Gadofosveset trisodium	Ablavar^®^ Vasovist^®^	0.25 M	0.03	0.12	19	Production discontinued due to poor sales.

**Table 3 jcm-13-02193-t003:** Classification of GBCAs in NSF risk classes (higher to lower) according to ESUR and ACR.

GBCA	ESUR	ACR *
Gadodiamide, linear (Omniscan^®^)	High-risk for NSF; suspended by EMA.	Group I: patients’ stratification based on eGFR required, contraindicated if CKD stage 4–5 and AKI.
Gadoversetamide, linear (Optimark^®^)	High-risk for NSF; suspended by EMA.	Group I: patients’ stratification based on eGFR required, contraindicated if CKD stage 4–5 and AKI.
Gadopentetic acid, linear (Magnevist^®^)	High-risk for NSF; suspended by EMA for intravenous use; only allowed for intra-articular administration in arthrography MRI.	Group I: patients’ stratification based on eGFR required, contraindicated if CKD stage 4–5 and AKI.
Gadobenic acid, linear (MultiHance^®^)	Intermediate-risk for NSF; approved by EMA for hepatobiliary imaging only.	Group II: recommended for patients with chronic kidney disease; assessment of renal function optional prior to intravenous administration.
Gadoxetic acid, linear (Eovist^®^, Primovist^®^)	Intermediate-risk for NSF; approved by EMA for hepatobiliary imaging only.	Group III (data regarding NSF risk remains limited despite an alternative hepatobiliary excretion pathway): patients’ stratification based on eGFR required.
Gadobutrol, cyclic (Gadavist^®^, Gadovist^®^)	Low-risk for NSF; assessment of renal function not mandatory prior to intravenous administration; caution in patients with eGFR < 30 mL/min (at least 7 days between two injections).	Group II: recommended for patients with chronic kidney disease; assessment of renal function optional prior to intravenous administration.
Gadoteridol, cyclic (Prohance^®^)	Low-risk for NSF; assessment of renal function not mandatory prior to intravenous administration; caution in patients with eGFR < 30 mL/min (at least 7 days between two injections).	Group II: recommended for patients with chronic kidney disease; assessment of renal function optional prior to intravenous administration.
Gadoteric acid (Dotarem^®^, Artirem^®^, Clariscan^®^)	Low-risk for NSF; assessment of renal function not mandatory prior to intravenous administration; caution in patients with eGFR < 30 mL/min (at least 7 days between two injections).	Group II: recommended for patients with chronic kidney disease; assessment of renal function optional prior to intravenous administration.

* Important note: ACR committee indications are less adherent to the more restrictive FDA guidelines, which recommend screening patients for AKI and conditions that may interfere with renal function, independently from the GBCA adopted.

**Table 4 jcm-13-02193-t004:** Summary of ESUR vs. ACR guidelines concerning GBCAs administration in pregnant and lactating women.

	ESUR	ACR
Pregnant women with preserved renal function	Smallest quantity of macrocyclic GBCAs only; very strong clinical indication to contrast enhanced MRI	Smallest quantity of macrocyclic GBCAs only; very strong clinical indication to contrast enhanced MRI
Pregnant women with impaired renal function	GBCAs formally contraindicated	Same safety measures as in non-pregnant women
Lactating women with preserved renal function	No breast-feeding interruption formally required	No breast-feeding interruption formally required
Lactating women with impaired renal function	GBCAs formally contraindicated	Same safety measures as in non-pregnant women

## Data Availability

Data are contained within the article.

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
