# Peer review of "Safe and Informed Use of Gadolinium-Based Contrast Agent in Body Magnetic Resonance Imaging: Where We Were and Where We Are"

_jcm, 2024, doi:10.3390/jcm13082193_

Round 1
Reviewer 1 Report
Comments and Suggestions for Authors
I enjoyed reading this review by Iacobellis et al. The manuscript contains a lot of interesting and well-structured information on the important topic of Gd contrast agents for MRI studies. The quality of the presentation and the literature scope are good. My only recommendation is the formatting of subscripted and superscripted parameters such as T1 and T2. The numbers (1 and 2) should be subscripted throughout the text. Also, please check for spaces between words. Otherwise, it should be published.
Reviewer 2 Report
Comments and Suggestions for Authors
Comments and Suggestions for Authors:
A review of the safe and aware use of gadolinium-based MRI contrast agents is presented in the manuscript. According to the authors, the goal of this review is to provide a summary of commercially available MRI contrast agents, their impact on imaging, and their negative effects on the body to aid in their appropriate selection for various clinical settings. Before its publication, the present work has several issues that should be addressed, as outlined below:
1. In the abstract 1,5 T should be corrected.
2. At the beginning of the introduction “free Gadolinium ion is highly toxic in vivo” add references for that.
3. The authors should add the structures of the MRI contrast agents based on their classifications.
4. To simplify, the authors should rephrase the following sentences on page 4: “Therefore, conversely to computed tomography (CT) iodinated contrast media in which at conventional doses it is well established the clinical superiority of non-ionic contrast media compared to ionic ones in terms of renal safety and adverse reactions, for GBCAs such difference between ionic and non-ionic molecules is far less relevant or even negligible in terms of their systemic osmotic effects due to the small volumes usually administered for clinical purposes”.
5. On page 5 in the section Extracellular Space Agents (ECSAs), the MRI agent's name should be added as examples like the following two sections.
6. Page 5 in section Hepatocyte-specific contrast agents (HSCAs): “they are preferentially up taken up by hepatocytes” should be rewritten.
7. Blood Pool Agents (BPAs): the only GBCA in this category is gadofosveset trisodium, that was discontinued after commercialization due to marketing reasons – Please add the reference for that.
8. Page 5 “this GBCA has approximately 5 times the relaxivity of ECSA, which allows first pass MR angiography to be performed with similar image quality as ECSAs but with a lower dose” – why Blood Pool Agents (BPAs) has approximately 5 times the relaxivity of ECSA.
9. Page 9. Rewrite this "Gadofosveset trisodium and BPAs".
10. Add reference for this ‘ However, at present Gadopiclenol has only been approved by FDA for clinical use in adults and pediatric patients aged 2 years and older; it has also been submitted to EMA for review, but (as of this writing, at the beginning of 2023) approval for marketing authorization is still pending.”
11. On pages 9 and 10 the author should make it clear what they mean by using 3’, 5’, 20’ and 45’ in the following sentence: “occurring between 3’ and 5’ from contrast injection” and “ about 20’ from injection with Gadoxetate disodium and about 45’ with Gadobenate dimeglumine for liver lesions depiction”.
12. Figure 4 uses the full form of LAVA.
13. Meaning of the “VI liver segment” in Figure 5.
14. Use the full formof STIR at least once.
15. Rewrite the following on page 13: “single shot fast spin echo heavily T2 weighted sequences (due to the T2 shortening effects of GBCA).”
16. Page 15 (according to EMA) guidelines): “)” should be removed.
17. Author should rewrite the frist paragraph of section 5.3. GBCAs and interaction with other drugs for better understanding.
Reviewer 3 Report
Comments and Suggestions for Authors
1. Oversimplification of classifications - Are the three categories of GBCAs an oversimplification, given complexities in behaviors and indications? Could additional subclasses be useful?
2. Limited focus on safety/risk - Does the review sufficiently examine safety issues like nephrogenic systemic fibrosis, long-term gadolinium retention risks, and differences between agents? Are the potential risks appropriately emphasized?
3. Incomplete dosing recommendations - Are all relevant dosing guidelines from regulatory agencies like EMA/FDA captured accurately for each agent? Could any dosing advice be ambiguous or misleading as written?
4. Lack of guidance on off-label use - Does the review properly advise that some uses discussed may be considered off-label in some jurisdictions? Is the status of recommended applications clearly stated?
5. Oversimplified optimization strategies - Are there caveats or complexities being overlooked in discussing timing of sequences or other optimization points? Could the strategies mislead some readers?
6. Insufficient referencing - Does the literature review appear to missing references to other important comparative studies or safety data on certain agents? Could the evidence base be more robust?
7. Readability for intended audience - Is the intended target readership of radiographers/technologists appropriately considered in terms of language complexity and level of assumed background knowledge?
8. Potential bias - Are any commercial considerations or conflicts of interest for the authors that could introduce potential for bias in how different agents are presented?
9. Outdated information - Given the review was published in 2024, is there a risk that some information on novel agents/indications may no longer be accurate due to advances in the last 2 years? Does it require an update?
Round 2
Reviewer 2 Report
Comments and Suggestions for Authors
I am satisfied with the author's response, and I have no additional concerns regarding the manuscript.
Reviewer 3 Report
Comments and Suggestions for Authors
Ok